# A Summary of Concepts, Procedures and Techniques Used by Forensic Entomologists and Proxies

**DOI:** 10.3390/insects14060536

**Published:** 2023-06-08

**Authors:** Tharindu B. Bambaradeniya, Paola A. Magni, Ian R. Dadour

**Affiliations:** 1School of Medical, Molecular & Forensic Sciences, Murdoch University, Murdoch, WA 6150, Australia; sritharindu27@gmail.com (T.B.B.); p.magni@murdoch.edu.au (P.A.M.); 2Harry Butler Institute, Murdoch University, Murdoch, WA 6150, Australia; 3Source Certain, P.O. Box 1570, Wangara, WA 6947, Australia

**Keywords:** casework, crime scene, skill base, entomology procedures

## Abstract

**Simple Summary:**

Forensic entomology is a globally accepted field of science that incorporates insect knowledge into crime scene investigations. It is crucial to update available concepts, procedures, and techniques of forensic entomology from time to time with new research findings for better use in the field and the laboratory. The current review provides a current account of field and laboratory guidelines for forensic entomologists and crime scene investigators/technicians to conduct entomological procedures related to casework.

**Abstract:**

Forensic entomology is a branch of forensic science that incorporates insects as a part of solving crime. Insect-based evidence recovered at a crime scene can be used to estimate the minimum postmortem interval, determine if a carcass/corpse has been relocated, and contribute to the cause and manner of death. The current review summarises the stepwise usage of forensic entomology methods at a crime scene and in the laboratory, including specimen collection and rearing, identification, xenobiotic detection, documentation, and referencing previous research and casework. It also provides three standards for the collection of insects when attending a crime scene. The Gold standard attributes to a forensic entomologist (FE) who is likely to be well-trained attending a scene. The subsequent standards (Silver and Bronze) have been added because the authors believe that this information is currently missing in the literature. The purpose is so that an attending crime scene agent/proxy with some basic knowledge and some simple tools can recover almost all the insect information required by an FE to make the best estimation of the minimum postmortem interval.

## 1. Introduction to Review

Although an update, this review is different to other publications which simply advise on the necessary processes and tools required by a fully trained forensic entomologist (FE) to conduct an entomological analysis of a crime scene [1,2]. In contrast to the conclusions of Hall et al., 2012 [1], that insect evidence should be collected and preserved either by an appropriately qualified person or under their instruction, the outcome of this paper is to provide an overview of how inexperienced personnel (with some training), such as crime scene technicians, forensic officers and proxies, attending a crime scene can sample and collect insect material, allowing the FE to make the best estimate of the time since death.

### Introduction to Forensic Science

Forensic science is a crucial constituent in crime-solving that agglutinates the investigation with the subsequent judicial activities [3,4]. Today, forensic science has diverged into numerous subfields; some of the more widely known sciences are forensic entomology, forensic toxicology, forensic anthropology, and forensic microbiology [4,5]. These disciplines provide principles and procedures to recover and analyse crime-associated evidence to reconstruct the actual circumstances of a crime scene [3]. This facilitates in the determination of a crime, which eventually results in a prosecution [6].

The physical evidence recovered from a crime scene is primarily used to verify offenders while exonerating any innocent persons, as well as how, when, and where the crime was committed [7]. The recovered physical evidence provides either observation-based clues or may need to be processed by advanced techniques to arrive at further conclusions. As an example, a microscopic analysis of hair could provide information such as the age, gender, and race of an individual present at a crime scene. This may be followed up with a DNA analysis and spectroscopic analysis, providing details relating to an individual’s identification and the presence of explosive and drug residues, respectively [8].

## 2. Forensic Entomology for Crime Scene Investigations

What is forensic entomology? It is known for using insects and other arthropods to aid in solving criminal and civil crimes [9,10]. Generally, forensic entomology involves three main subcategories: urban, stored product, and medicolegal entomology [8]. Urban entomology investigates the insect-related structural defects of buildings and medical myiasis [11,12]. Stored product entomology is used to assess biosecurity risks associated with food contamination and environmental pollution [13]. The last category, medicolegal entomology, uses insect-based evidence to investigate numerous criminal activities associated with homicides, suspicious fatalities, abuse of vulnerable people, hospital neglect, animal cruelty, contraband trafficking, and wildlife poaching [9,10,14].

The primary function of insects in crime solving is as biological indicators to estimate the time elapsed since death or the minimum postmortem interval (minPMI) [15]. Furthermore, the colonisation time of dipteran larvae in myiasis wounds can provide an indication of the time of neglect for humans, pets, and livestock [16]. The two fundamental principles of utilising insects are determining the minPMI based on their predictable temperature-based development and the sequential colonizing patterns on carcasses [9]. Flies (Order Diptera) are the main protagonists of all the carrion arthropods that are predominantly considered for use in determining the postmortem interval due to their involvement in carcass decomposition as both early and enduring colonisers [17,18]. The role of flies in carcass decomposition interchanges between scavengers of decaying tissues and prey to other predacious invertebrates (e.g., beetles and ants) and vertebrates (e.g., hyenas and vultures) [18,19] (Figure 1a–d).

Typically, the minPMI for early stages of carcass decomposition (i.e., 3–72 h following death) is estimated by pathological methods [20]. However, after this period, fly eggs, larval, and pupal specimens become one of the most reliable indices for the age determination of a carcass [21,22]. This is achieved by three methods: (1) calculating the accumulated degree day/hour units, (2) referring to life history tables, and (3) using species-specific isomegalen/isomorphen curves [23]. In addition, the minPMI in advanced stages of decomposition can be estimated by assessing the succession patterns of flies and other carrion-associated invertebrates, although this is generally less accurate [24]. However, estimating the minPMI based on fly development in advanced stages of decomposition has some drawbacks, for example, the fly larvae and full puparia sampled may not represent the earliest colonisers, which may have already completed their life cycle [25,26]. Generally, the predicted sequence of attracting flies to carrion begins with the Calliphoridae, followed by Sarcophagidae, Muscidae, Sphaeroceridae, Piophilidae, Fannidae, and Phoridae [9,23]. Carcasses are also colonised by beetles (Staphylinidae, Scarabaeidae, Carabidae, Silphidae, Cleridae, and Dermestidae); moths (Pyralidae and Tineidae); and mites (Pterygosomatidae and Omentolaelapidae), which can also be forensically significant [27,28]. Furthermore, the sequence of these invertebrates invading a carcass can vary based on external and internal influences, such as changes in the carcass characteristics (size, nutrient composition, existing decomposition stage, and xenobiotics), abiotic (temperature and humidity), and biotic (terrestrial and aquatic habitats) factors [26].

Apart from estimating the minPMI, entomological assessments in some cases can be made to determine the transfer of carcasses from their original crime locations [29]. This is underpinned by the concept that the entomofauna of a carcass resembles the overall insect diversity of the environment where it originally resided, and any secondary movement to an alternative place could change this crime location-specific entomofauna [30,31]. This is generally context driven, depending on the crime scene location and environmental conditions. In addition, disturbance to a carcass may cause any fly larval aggregations to disperse, which are especially visible when densely packed larval masses are present [9]. Additionally, there is a potential use of the characterised internal and external microbial profiles of larval masses infesting carcasses to verify the originality of a crime scene. This could be achieved by assessing the homogeneity of microbial profiles between the larval masses and the crime scene [32].

It has been widely demonstrated that fly tissues can be potentially used for victim identification and determining the cause of death under entomogenetics and entomotoxicology [33,34]. This is crucial when the tissues and fluids of victims are not available to recover from the body for analysis due to the occurrence of tissue degradation by larval feeding, scavenging, and burning [35,36]. An analysis of the contents of parts of the fly alimentary tract using DNA may also reveal the identity of a victim [37]. To identify fly species, DNA can also be extracted from flies at various stages of life, followed by sequencing species-specific nucleotides of their genes, and then referring these to a genetic database [33,34]. Moreover, the age estimation of larvae and pupae based on gene expression is another molecular approach [38].

Previous entomotoxicological studies have shown that fly tissues can be deployed to detect alcohol (e.g., ethanol), drugs (e.g., antidepressants, barbiturates, benzodiazepines, opioids, and phenothiazine), metals (e.g., thioridazine, antimony, barium, cadmium, lead, and mercury), and pesticides (e.g., malathion, parathion) [39,40]. However, the success of these genetic and toxicological determinations using flies depends on the species, development stages, feeding activities, environmental conditions, insect sampling techniques, specimen number, sampling frequency, and insect killing and preserving method [40].

### 2.1. Forensic Entomologist: An Apprentice to an Expert

The analyses of entomofauna at a crime scene is complex, involving sampling and identifying insects, determining species–habitat relationships, evaluating the development stages and successional patterns, and assessing insect ethology concerning variations in climatic parameters [41]. Becoming a practitioner in forensic entomology requires training in the fundamentals of entomology and ecology. To conduct these tasks, an FE must understand concepts such as insect physiology, taxonomic and phylogenetic relationships, the behavioural adaptation of insects for different ecological setups, and their impacts on human affairs [42,43]. This training should then be narrowed down to specific topics that directly address the diagnostic characteristics and interrelationships of forensically important insect groups. This subset of knowledge related to forensic entomology can be acquired by following an undergraduate or postgraduate level specialised course or a degree program [44]. In most countries, these academic qualifications provide the required accreditation and recognition for an FE to work with their local law enforcement agencies [9,44]. Please note that any academic pursuit must go hand in glove with the understanding and practical aspects of crime scene attendance and protocols, as well as the preparation of expert witness statements and the presentation of such evidence to the court.

These skills and experiences can be developed by conducting mock crime scene trials utilizing human surrogate carcasses (e.g., swine) [43,45]. It is best practice to conduct these field studies with an FE who has experience in actual casework [43,45]. In addition, it is essential to acquire background knowledge of other related forensic subfields such as molecular biology, microbiology, taphonomy, toxicology, pathology, and criminalistics that may be integrated with forensic entomology [44,46].

### 2.2. Forensic Entomology in Crime Investigation: Complexity and Constraints

Insects play a pivotal role in carcass decomposition [47]. Flies and other arthropods visit a carcass if there are no physical barriers to block their access. When the food source (carcass) is exhausted, the insects leave [48]. The insect colonisation of a carcass primarily relies on the seasonal- and regional-based climate, including temperature, humidity, and photoperiod. It also relies on the intra and interspecies dynamics, including feeding preferences and competition and various habitat scenarios such as terrestrial and aquatic, urban and rural, indoor and outdoor, and burial and exposed [26]. In addition, secondary factors such as the method of death, including burning, poisoning, drowning, dismemberment, concealment, and hanging, as well as carcass features such as age, size, clothing, and trauma also influence the patterns of insect colonisation while impacting on the decomposition process. This complexity associated with insect and carcass interactions consistently challenges an FE to conduct their tasks diligently and accurately. However, knowledge of local entomofauna, prior experience participating in insect-related casework, and the skills in incorporating research into casework are the building blocks required to best estimate the minPMI [49].

Besides these carcass insect complexities, an FE must also deal with many regulatory challenges such as interdisciplinary cooperation with other investigators at crime scenes, including other scientists and forensic field agents, strict deadlines to submit expert witness statements, and courtroom conflicts [50,51].

### 2.3. Progression and Standardisation in Forensic Entomology

Several early classic works by Mégnin (1894) [52] and Payne (1965) [53] highlighted the importance of insects in contributing to the carrion decomposition process. These studies laid the conceptual framework to utilise insects in solving crime. Mégnin (1894) [52] proposed the first formal definition and testable mechanism of insect succession in carrion, and Payne (1965) [53] emphasised the significant involvement of insects in reducing carrion biomass in the early stages of decomposition. Following these formative works, many studies have been conducted in different regions of the world to introduce identification keys for carrion insects, document carrion-associated insect succession under different temporal and spatial conditions, record fly development periods under field and laboratory conditions, and unite forensic entomology with other related fields such as genetics, toxicology, and microbiology [15,54]. These findings have been incorporated into many crime investigations, published as casework in books and scientific journals, presented in conference proceedings, and sometimes proffered by popular media. All have contributed to the establishment of forensic entomology as a reliable subfield of forensic science [54].

The standardisation of entomological methods developed for solving crime has strengthened the science of forensic entomology and has allowed this science to gain more visibility and acceptability among law enforcement agencies. Several crucial studies have been published recently that critically evaluate many successional and development studies, bridging the gap between research and casework. This has led to the improvement of standards and protocols for the FE to use in real crime situations [23,55,56,57].

Numerous attempts have been made to propose global standards to assist an FE in performing their work logically and systematically under field and laboratory conditions [2,9,56,58]. These standards outline the basic procedures and the equipment required to conduct entomology-associated tasks by an FE at a crime scene and in a laboratory to generate accurate estimates of the minPMI [9,58]. However, these standards are not rigid and can be modified (Table 1) based on the degree of training and experience of the practitioner, availability of resources, and the circumstances of the crime scene. The three standards presented here (Gold, Silver, and Bronze) do not substantially change the basic procedures but provide some alternative ways to limit any misinterpretations [9,49]. At times, an FE cannot attend a crime scene to perform the standard entomology procedures (Gold standards) and it is left to a proxy (e.g., pathologist, crime scene technician or a law enforcement agent) with limited entomological expertise to collect the insect material. The alternative methods (Silver and Bronze standards) are for these proxies. These representatives have different degrees of knowledge, ranging from direct training from an FE and prior experience assisting an FE at a crime scene (Silver standard) to minimal or zero exposure to forensic entomology (Bronze standard). Although they are by no means perfect, these procedures may still be adequate for an FE to construct a reasonable estimate of the minPMI. This evidence may not serve as court evidence but simply contribute to “collar and cuffing evidence”, giving investigators a time frame to include or exclude people associated with the crime. To assist in this endeavour, a quick study of a standard operating procedure concerning the collection of insect evidence or viewing an entomological crime scene application [59] may be appropriate. Such standards should be revised in a timely manner to upgrade them with novel concepts and techniques. In the next section of this review, a crime scene technician or proxy may find the following information useful and set the context for them as to the underlying requirements of the FE for calculating the minPMI.

## 3. Crime Scene and Health Pathology Facility

The following procedures fall into two major categories based on the location where specific entomological interpretations are undertaken: crime scene/pathology facility and the entomology laboratory [1,9].

Generally, FEs or proxies collect insect evidence at a crime scene, or later at a health pathology facility, where autopsies are usually conducted. The remains are assigned to these facilities by police, pathologists, and medical examiners/coroners. A request then follows if insect material is evident to determine the minPMI and/or provide a general statement about the entomological context of the crime scene [10,49].

At any of these sites, an FE or proxy must perform several standard tasks (Table 1), namely, the collection of the crime scene parameters (being present or via video and photos), including climatic parameters, insects, and their remnants (e.g., mouth parts, appendages, wings, empty puparia). Following collection, some insects may be preserved and stored, and some placed (40% of eggs larvae and full puparia) on a medium to be reared, followed by transporting these live and dead specimens to an entomology laboratory for further analysis [1,2,9].

An FE or proxy should adhere to the general guidelines set by the law enforcement agency controlling the crime scene [49]. An FE or proxy, similar to all other investigators at a crime scene, must wear the appropriate clothing and personal protection equipment (e.g., closed-toed shoes, long pants, scrubs, (disposable overalls) and masks) to avoid any possible contamination and biohazard risks [9,49]. They must visit the scenes with the necessary equipment and documentation to fulfil three major purposes. Firstly, collecting insects (live and/or dead): labelled plastic containers and vials, a handheld lens, hot water (insulated bottle), forceps, fined tipped artist paint brushes, an insect net, bedding materials (e.g., dry sand, sawdust, vermiculite), and food sources (e.g., beef, pork pieces) for rearing purposes, and ethanol (ethyl alcohol) for killing and preserving insects. Secondly, gather the microclimatic data: a portable weather gauge, an infrared thermometer, and a temperature data lodging device. Thirdly, document necessary information via prepared data entry sheets. An FE should have an a priori knowledge of the standard operating procedures of law enforcement at a particular location and this should determine if other devices such as a camera and a video recorder should be included equipment [9,49].

### 3.1. Collection of Onsite Data

The collection and preservation of entomological evidence at a crime scene should be carried out as soon as possible after a body is discovered [60]. The delayed recording and sampling of entomofauna could potentially generate misleading information. This is due to the development, succession, and orientation patterns of carrion insects, which continuously alter with climatic changes and other interruptions caused by inadvertent destruction by investigator activities and scavenging of the remains by other vertebrates and invertebrates [61].

Due to a possible high risk of damage and contamination, the insect evidence should be safeguarded by the law enforcement agency protocols at a crime scene. When law enforcement establishes a crime scene perimeter, it is essential to consider the area where insect activity may be evident in and around a carcass [62]. This is dependent on the decomposition stage and the age of the immature insects on the remains [49]. Specifically, eggs and feeding larval stages of flies are restricted to the carcass; however, post-feeding larval and pupal stages can be dispersed underneath the carcass, within the surrounding cadaveric soil, some distance from the remains and buried in the soil. In dwellings, these stages maybe located near physical objectives present in the vicinity of the carcass (e.g., under carpets, cupboards, and other household items) [1,9]. In addition, the dispersion of adult flies inside dwellings could also be influenced by nearby fly attraction sources, such as lights, windows, and mirrors [48]. Although not always possible, the best time to visit an outdoor crime scene to conduct an entomofauna investigation is in the daytime. Firstly, the perspective is clearer of the area surrounding the crime scene, and secondly, insect fauna is mostly diurnal. Undesirably, due to the utilisation of light sources at night around a crime scene, many other extraneous insects may be attracted to the remains compromising any typical insect activity surrounding the carcass [53].

It is essential to request from the forensic investigator in charge any related information concerning the management of the scene. This should include photographs and video footage from any initial visits of investigators to the scene, and any evidence of physical alterations made to the immediate environment after the remains were discovered (e.g., opening windows, switching on/off heaters or air conditioners, and placing light sources to conduct investigations at night). A potential bias associated with a crime scene is informing the FE or proxy about the date of the last sighting of the deceased. Such information should ideally only be obtained after estimations of the minPMI have been ascertained by the FE [51].

Following the arrival at a crime scene and consulting with law enforcement, the FE or proxy should be able to view the remains and observe the patterns of insects on and around the carcass. Following this assessment, the FE or proxy can realise what tools are required to work the scene. Sometimes, large larval aggregations may be present in and around the natural orifices of carcasses, such as in and around the mouth, nostrils, ears, and the genital region [63]. These body regions provide protection and suitable soft tissues for larval feeding in the early stages of development [63,64]. These early colonizing sites can also occur on other parts of the body based on factors such as surface humidity, insulation, body orientation, clothing, and the wrapping of bodies and wounds [64]. For instance, when bodies are hung above the ground, decomposition fluids and larvae fall on the ground directly beneath the body, known as the ‘drip zone’ [65]. In addition, when bodies are enclosed in suitcases, fly egg clusters may be visible on the exposed zips of these suitcases [66]. These observations of early fly oviposition and larval aggregations may provide useful insights on antemortem determinations, such as the cause and manner of death [64].

The study of carrion decomposition is also an integral part of the basic training to become an FE, and typically involves the determination of the decomposition stage of a cadaver [61]. The insect colonizing patterns associated with different decomposition stages can provide clues to determine the minPMI [67]. Generally, on land, carcass decomposition comprises five main stages: fresh, bloated, active decay, advanced decay, and dry/remain stages [68]. In contrast, in water, this is generally six stages: submerged fresh, early floating, floating decay, bloated deterioration, floating remains, and sunken remains [69]. When assuming an existing decomposition stage, the estimated time to reach that stage should then be calculated in conjunction with specific factors, which directly modify the chemical composition of the carcass and in turn reflect the insect colonisation patterns. These factors include the climatic parameters (temperature, humidity, and photoperiod), the fly access (concealment and inside dwellings), and how the decedent died (e.g., hanging, burning, and drowning). Typically, hanging bodies require sampling from both the corpse and from the drip zone, if it has established on the soil directly below the corpse. A light burn might enhance the body for colonisation as the skin splits, but a more severe burn repels insects [9]. When a carcass is submerged in a body of water, it reduces terrestrial faunal attraction while favouring access by aquatic fauna. In addition, size and age of the carcass impacts the constitution and amount of flesh and body liquids affecting insect colonisation and decomposition [26]. Additionally, it is vital to record any other sources nearby that may impact insect attraction to the carcass (garbage bins, toilets, waste management sites, and another carcass). This could significantly influence the insect abundance and successional patterns [9].

### 3.2. Collection of Microclimatic Data at a Crime Scene

Insects are poikilothermic, their body temperatures fluctuate along with the ambient temperatures to be in homeostasis [70]. Therefore, entomological assessments at crime scenes should be conducted concerning ambient temperatures, as it affects the development and succession patterns of carrion insects [71,72]. Specifically, temperature impacts insect development by regulating their movement, metamorphosis, reproduction, and diapause [73]. In contrast, succession patterns change with temperature fluctuations as it affects the microbial activities associated with the carcass decomposition [74].

The best available source to obtain environmental temperature readings at a crime location is the crime scene itself matched to the temperatures from the nearest meteorological station [2]. Investigators can request daily or hourly temperature data for as many days as required prior to discovering the remains [2,9,75]. However, when cadavers are inside a dwelling, these meteorological station data are only helpful if the indoor space is open to the outdoor by windows, doors, and vents. In such situations, the outside temperature data must be paired with the indoor temperatures when constructing the thermal simulation model for the crime scene environment [76]. In situations where the indoor temperature is controlled (heaters, fans, and air conditioners), the temperature produced by these devices should be reviewed as part of the temperature data. These temperatures must also be correlated with the actual temperature readings at the site [9]. At the crime scene, hourly or minimum/maximum temperatures can be measured by a temperature data logging device placed near the carcass, protected with a Stevenson screen [49]. It is suggested that meteorological data be optimised by conducting a site-specific error calculation via a regression analysis to estimate the minPMI [77,78].

Apart from the environmental temperatures, body, water, soil, and larval mass temperatures should also be recorded based on the context of the crime. A contactless infrared thermometer can measure the body, and larval mass temperatures, and the soil and water body temperatures can be recorded by a temperature probe [9,49]. The decision of the temperature data most relevant for minPMI estimations must be made based on their degree of heat transfer for larval development. For example, it is necessary to consider aggregated larval mass (maggot mass) temperature for estimating the minPMI. As a rule, large larval masses can be up to 20 °C above ambient temperature, so that when larvae are identified as being part of a larval mass, this heat source is then the most relevant to late second instar and third instar larval growth, not the ambient temperature [23].

In addition, it is vital to record extreme weather events, such as heavy rain within the period of interest, as these events could also affect the arrival, development, and succession patterns of carrion insects [79]. Humidity and photoperiod can also be recorded along with temperature. Moon phase is also important, as a waxing gibbous moon may affect adult blow fly movement. Such data are vital when recreating crime scene conditions, or a research study in a growth chamber to determine the development rate of larvae [80].

### 3.3. Recovery of Insects for Storage or Rearing Purposes

The specimens collected at a crime scene should be a clear cross-section of the different development stages (eggs, larvae, full and empty puparia, and adults) of various insect successional groups (flies, beetles, mites, and moths) wherever located on the carcass and its surrounding environment [2]. The number of individuals recovered at each stage of development must be adequate for conducting a range of entomological assessments in the laboratory (i.e., species identification, PMI estimation, and xenobiotic determination). However, these sampling numbers depend on their availability and could range from a single egg or larva to several egg rafts and larval masses [81].

Insects collected at a crime scene should be divided into two batches, one batch preserved and the other reared to adult. This is generally performed at the scene, but if appropriate equipment is unavailable, then they must be cooled (not frozen) to slow down development and dispatched rapidly to a laboratory [9]. Preserved specimens are used for identifying the species, determining their age at the time of sampling based on their instar stage, the larval length and width measurement, and extracting gut contents for genetic and toxicological analysis [82].

#### 3.3.1. Preservation of Insects

As mentioned above, insects recovered from a carcass should be preserved at their collection point [9]. When kept alive in containers to preserve for a later time and not cooled, their age status will alter, may exacerbate some cannibalistic behaviour, and reduce the number of collected larvae for further analyses [83]. In such circumstances, the development rate is compromised, and such samples should not be used to estimate the minPMI [83]. However, if a complete record of the thermal history of a sample is provided, an FE should be able to backtrack the age of the insects at the collection time and extrapolate useful information concerning the minPMI for the investigators.

Fly eggs and adult stages can be difficult for estimating the PMI, as limited knowledge is available pertaining to only a few species [9,84]. However, sometimes, the only samples collected are these life history stages, and a practiced FE may be able to identify the morphological changes that occur during egg embryogenesis [78], as well as using some of the methods described by Tyndale-Biscoe (1984) [85] with regard to the age of adults, especially those captured inside a dwelling. Once eggs are located, they can be collected after differentiating them from soil and litter particles using a handheld lens and a fined-tipped artist paint brush [86]. In contrast, the adult flies at the scene can be trapped using an insect net. Following capture, they can simply be removed by dipping or spraying the fly with 70% alcohol and then transferring the dead specimen to a vial. If alcohol is unavailable, then the adult fly can be captured in the net and carefully placed into an insect killing jar prior to preservation [9]. Alternatively, insect sticky paper can be used during daylight to passively collect adult flies [87]. Following their collection, individual flies can be extracted from the sticky paper using a fine-tipped paintbrush and vegetable oil.

When collecting delicate first and second instar larvae, a fine-tipped paintbrush can be used. Older pre- and post-third instar larvae and pupal stages can be removed by forceps or a spoon (i.e., designed to fit inside a collection vial) [9,49]. Unlike forceps, which are limited to collecting individual larva, a spoon can easily collect a large cross-section of larvae across a surface or inside a larval mass [49]. Collected larvae should then be placed in hot water (not boiling) for approximately 1 min before being placed in vials containing 80% ethanol [88]. In order to make sure that the morphological features of the insect in metamorphosis are preserved, full puparia should be pierced with a pin before being placed in hot water, followed by immersion into 70–80% ethanol [89,90]. The required hot water can be carried to the scene in a thermos flask/cup or an insulated bottle and can sometimes be obtained from a café or take-away establishment nearby. Prior to storing these vials, an appropriate label should be placed on the outside (written in indelible ink) indicating the time, date, sampling location, person collecting, and the part of body where sample was collected. A second label with the same information but written in pencil (pencil will not dissolve in alcohol) should be inserted into the vial [2,9].

Beetles collected at a crime scene should be handpicked using forceps and hot water, dipped to kill them before being placed in vials containing 70% ethanol. Beetles may be observed by thoroughly examining the litter surrounding the carcass as well as sieving the soil under the carcass to a depth of approximately 20 cm [91]. It is worthwhile to mention here that Silver and Bronze personnel should collect all insects associated with the corpse and allow the FE to make the necessary decisions on their relevance.

When dealing with a crime scene located in an aquatic environment, the larval stages of freshwater insects such as mayflies (Order: Ephemeroptera), stone flies (Order: Plecoptera), and caddis flies (Order: Trichoptera) should be collected from submerged and floating bodies into plastic vials (70% ethanol) using forceps. Once again, it is recommended that Silver and Bronze personnel collect all aquatic fauna that they observe associated with the corpse. In contrast, it is recommended to preserve approximately 40% of sampled specimens from stream-bottom remains in properly labelled vials containing 95% ethanol to minimise their decay during storage [71,92].

The remaining live aquatic insects collected from a carcass should be transported to the laboratory for rearing and identification. These aquatic insects should be placed in plastic containers with water from where they were sampled (e.g., a lake, sea, pond, or stream). During transport, the specimens should be kept cool by placing the container in ice and the container lid should be opened occasionally to aerate the water [69].

#### 3.3.2. Preparation of Insects at the Crime Scene for Rearing Purposes

The live insects for rearing purposes collected at the crime scene should be placed on a dish (Petri dish, foil) with a suitable food source (e.g., meat). This dish is then placed on 2–3 cm of bedding material (sawdust, vermiculite, builders’ sand) inside a container with a mesh lid [9]. When preparing full puparia for transporting to the laboratory, they can be directly transferred into containers half filled with bedding material with a standard lid. No feeding source or ventilation is required if all insect material is transported inside a cooler or kept refrigerated.

### 3.4. Laboratory

The following sections pertain to the work that an FE must conduct on either the material collected by themselves, or the material collected by their proxy. To identify, measure larval, pupal, and adult external body characters and rear immatures to adult stage. An entomology laboratory should be equipped with the following basic apparatus: stereo microscopes, vernier callipers, growth chambers or temperature-regulated rooms, preservative chemicals (ethanol), gloves, forceps, scalpels, and Petri dishes. Other equipment may include DNA/RNA reagent kits, PCR apparatus and genetic sequencers to identify insects [93]. Please note that carrion insects, brought into an entomology laboratory, either alive or dead, should be handled with care to enable a range of analyses to be conducted [9].

### 3.5. Identification

The identification of specimens to species level should be completed first. Without this knowledge, other analyses will be compromised [94]. Generally, insect identifications are made using morphological and/or molecular methods. Other techniques such as hyperspectral or CT imaging, tomography, and chemical techniques may also be used; however, each has associated pros and cons according to expense and availability [95,96]. Nonetheless, whichever technique is used, it should provide the best identification outcome. In fact, the most informative and best identification is to send the collected specimens relevant to the minPMI to a dipteran taxonomist. The following sections discuss the two main techniques used to identify insects: the morphological identification and molecular identification.

The morphological identification of carrion insects to species level can be achieved using taxonomic keys and standard or electron microscopic images [97,98]. Dichotomous taxonomic keys provide a stepwise assessment using morphological characters that guide the user to identify the species [94]. The level of accuracy and confidence in using taxonomic keys for species determination develops with a skill base in entomology and long-term usage to resolve different morphological characters of specimens through microscopic examination [94]. Comprehensive keys for fly identification are numerous and have been generated by James (1947) [99] and Zumpt (1965) [100] and later upgraded by Smith (1986) [101] and Wood (1989) [102]. Recently, De Carvalho et al. (2008) [103], Szpila et al. (2015) [104], Whitworth (2019) [105], and Wallman (2001) [106] proposed identification keys for flies, focusing on specific species, development stages, and their geographical locations. In contrast, fewer taxonomic keys exist for forensically important beetles [107,108].

Ultrastructure studies of eggs, larvae, full and empty puparia, and adult stages of different fly species by scanning electron microscopes (SEM) aid in precise species identification. Typical egg-associated ultrastructures are micropylar plate, median area, respiratory plastron, hatching lines, and chorion sculpturing [109,110]. The suite of larval characters includes the posterior spiracles, respiratory slits, peritreme ornamentation, anterior spiracles, cephaloskeleton, and spine arrangement [110,111]. When identifying full puparia, characteristics such as bubble membranes and respiratory horns are used, along with spine arrangements and posterior spiracles that are correlated with their preceding larval structures [96,108,109,110]. The typical characteristics for the species distinction of adult stages include setae pattern arrangement on the thorax and wing venation [112].

In addition, some SEM-based ultrastructures of forensically important coleopteran have been published for several families such as the Nitidulide and Carabidae. These ultrastructures include the antennomeres, head capsulae, and mouthparts [113,114].

The incorporation of molecular methods, specifically DNA barcoding for identifying crime-associated insects, is an alternative if the user does not have appropriate taxonomic training but has the necessary laboratory skills or can outsource to a molecular biology laboratory to perform the molecular techniques [115]. Additionally, in situations where fewer specimens or fragments of specimens with no diagnostic characters are recovered and where there is an unavailability of taxonomic keys to refer to for the collected development stage, molecular methods are the best recourse for species identification [115,116,117].

DNA barcoding for insect identification is directed to detect a unique short sequence of nucleotides of the genome of each species to discriminate from other related species [118]. It is widely accepted that Cytochrome C oxidase subunit 1 (CO1), the last enzyme of the mitochondrial respiratory chain, is a suitable marker for the DNA barcoding of insects due to its slow mutation rate and highly conserved nature across species [119].

Generally, molecular identification of crime-collected specimens are conducted under five essential steps: preparation of samples, DNA extraction, PCR amplification, sequencing, and sequence library identification [118]. Adult, larval, and pupal stages of flies stored in 70% ethanol can be used for DNA extraction purposes. The specific primers for PCR amplification can be obtained by referring to previous publications [35,36]. However, limited genetic studies conducted on forensic flies in different locations of the world means that there is a scarcity of the sequence data of those species in these regions [117].

Previous studies that have provided the DNA sequences for carrion dipterans include Stevens and Wall (2001) [120] for Calliphorid species in the United Kingdom, Wallman et al. (2005) [121] for Calliphorid species in Australia, Harvey et al. (2003) [122] for *Calliphora dubia* (Macquart), *Chrysomya rufifacies* (Maquart), and *Lucilia cuprina* (Meigen), Harvey et al. (2003) [123] for Calliphorid species in Southern Africa and Australia, and the global scale study on Calliphorid species by Harvey et al. (2008) [124].

### 3.6. Examining Larvae and Full Puparia for Age Determination

Development studies provide time frames required for immature flies that can be used as indications to estimate the minPMI [23]. The instar of fly larvae can be determined by a microscopic observation of the number of respiratory slits available at their posterior spiracles [100]. Typically, a first instar blow fly larva has a single slit, whereas second and third instar larvae have two and three slits, respectively. In contrast, pre- and post-feeding stages of third instar larvae are discriminated by their body colour change (translucent to opaque), a food distended crop, body compaction, and wandering behaviour away from the carcass [23].

In addition, the length and width of larvae can also be considered as an indication of age [125]. The length and width of larvae can be measured by a stage micrometre and a vernier calliper. The correlation of actual length and width data of a sampled individual with reference values given in previous publications aid in determining the age of the specimen, hence the PMI. However, temperature variations, the type of food substrates, and the presence of xenobiotics in it reflect on any of these age determinations. Magni et al. (2012) [117] referred to the length of oven-dried larvae of *Lucilia illustris* (Meigen) with desiccated larval specimens collected at a crime scene for their age determination. Some of the essential studies with length change reference values are available for *Ch. megacephala* [126,127,128], *Ch. rufifacies* [129,130], *Ch. bezziana* [131], *L. sericata* [132], and *L. cuprina* [133].

Furthermore, in previous studies, length and instar stage variation of larvae under different constant temperature regimes were demonstrated using isomegalen and isomorphen curves, respectively. These curves are available for species such as, *Calliphora vicina* (Robineae-Desvoidy), *L. sericata*, *Ch. albiceps*, *Protophormia terraenovae* (Robineae-Desvoidy), *Ch. megacephala*, and *Liopygia argyrostoma* (Robineae-Desvoidy) [128,132,133].

Pupal morphogenesis can be considered for minPMI estimations as the full puparia coexist with carcasses over a prolonged period [89,134]. The extent of the external organ development of pupae can be examined via a stereo microscope after removing the puparium using a surgical knife and forceps. A photograph showing the status of the external organ development of a sampled pupa can be coupled with a reference photograph series given for the same species in a previous publication for the age determination. However, it is essential to consider the temperature variations of the environment that the full puparia were exposed to prior to sampling, as low temperatures can delay pupal morphogenesis [130]. In this regard, photographic timelines for external organ developments of pupae (intra-puparium development) are given for *Megaselia spiracularis* (Schmitz) [135], *Megaselia scalaris* (Loew) [136], *Calliphora vicina* (Rob-Desvoidy) [90], *Hermetia illucens* (L.) [137], *Ch. Megacephala* [138], *Ch. rufifacies*, and *Dohrniphora cornuta* (Bigot) [139].

Insect aging of larvae and full puparia can also be performed in the laboratory using methods such as hyperspectral [125] or CT imaging [140], tomography [141], and biomolecular [142] and chemical [143] techniques. Such techniques require advanced instrumentation and a high level of expertise to conduct these analyses.

### 3.7. Rearing

The rearing of eggs, larval, and pupal specimens until the adult stage can often facilitate a more accurate species identification [144]. Adult-level identification has a greater certainty in correct species determination, and they are well-described in taxonomic keys [145]. When rearing dipteran species in an insectary, they should be confined in insect cages and provided ad libitum with food, water, and oviposition substrates [17,23].

Immature stages are best reared in growth chambers whereby temperature, humidity, and photoperiod can be controlled. Such development studies are used to determine the time duration required to ascertain the development stage of the sampled species and their larval length and width changes. It is recommended to use swine muscle as larval food for these growth chamber studies as the domestic pig (*Sus scrofa* L.) is primarily considered as the proxy of humans when conducting development studies [146]. Additionally, when dealing with a wildlife-associated crime, the identical tissue type should be the feeding substrate utilised in growth chamber studies [147].

### 3.8. Xenobiotic Detection

The utilisation of insects for xenobiotic detection relies on the prolonged tissue retention of drugs, pesticides, and metals, leading to high sensitivity for analytical detection techniques [39,148].

The fly larval, pupal, and adult stages collected at a crime scene can be used as a source to extract and detect chemicals. Previous studies showed that larvae collected from internal organs such as liver, or from the head and muscles of a carcass retain high concentrations of chemicals [40].

The larvae sourced from these regions should be kept alive and starved for several hours prior to analysis to ensure the absorption of any chemicals into their metabolic system. It is recommended to store killed larvae under dry conditions at −20 °C. Such steps will ensure a higher retention and concentration of chemicals because storing in ethanol may diminish trace volumes of drugs in larvae [40,149].

At the laboratory, sampled fly development stages can be prepared for analysis via three methods: (1) macerated and homogenised, (2) digested via a strong acid or enzyme, and (3) pulverised by grinding. The available analytical methods for these xenobiotic detections are immunoassay, high-performance liquid chromatography (HPLC), liquid chromatography-mass spectrometry (LC-MS), and gas chromatography-mass spectrometry (GC-MS) [30].

## 4. Documentation

Documentation at a crime scene is vital to conserve the integrity of the evidence and for the continuity of the chain of custody. Previous studies showed that the layout of the protocol sheets used by an FE varies because it is based on the requirements of investigation agencies and legal systems in that region [150]. This documentation concerns two parts [151]. Firstly, the crime scene processes (information gathered by either the FE or proxy) and secondly the laboratory processes (information processed by the FE only). Once these processes are complete, the FE should have enough information to produce an expert witness statement [152] (Table 2).

Generally, an FE will visit the crime scene once; therefore, it is necessary to record all the information while present at a crime scene [9]. Occasionally, the FE may have to return to the scene to gain some context surrounding the scene as well as sample more of the insect material because the original visit was at night. To retain more time for sampling while keeping documentation to a minimum, it is best practice to have a protocol sheet containing specific questions with as many possible multiple-choice answers. These sheets should be designed to gather all essential information related to a crime scene that explains the crime characteristics, microclimatic data, quantity and quality of sampling specimens, and the details of the sampler. Additionally, the laboratory data sheet should be prepared to document all the analyses conducted in the laboratory, which an expert can easily refer to for verification purposes [60].

### 4.1. Referring to the Literature

The accuracy of establishing a successful entomological interpretation of the circumstances at a crime scene depends on the empirical analysis developed by an FE throughout their career [44]. One of the best ways of acquiring and upgrading the knowledge and skills associated with insects for solving crime is referring to previously published research and casework [153]. These publications essentially guide a practitioner to incorporate the available knowledge into procedures and provide a comprehensive guideline for researchers and academics to develop training models and conduct future research [153].

### 4.2. Fly Development Studies

Development studies outline the variations in the fly life history [23]. The data from these development studies can be directly incorporated to estimate the minPMI, as these publications essentially contain life history tables and isomegalen/isomorphen curves. The timeline data given in life history tables and the lower development threshold temperatures can be used to calculate the ADD/ADH [25,154], and hence, the minPMI. However, when selecting a previous development study to corroborate a minPMI estimation of a particular fly species, the FE should determine if the temperature regimes, photoperiod and humidity, and larval feeding tissue type, along with the location where the source colonies originated, are appropriate to use [155].

Besides providing the baseline data required to calculate the minPMI, these studies highlight the laboratory protocols that must be maintained when conducting trials related to crime scenes. These protocols need to detail the colony generation, rearing and maintaining the genetic diversity of the colony, simulation of environmental conditions in an insectary/growth chamber, and larval sampling and data gathering [17]. Numerous authors have conducted development studies on numerous blow fly species (Table 3).

### 4.3. Insect Successional Studies

Insect successional studies provide a descriptive account of insect colonisation patterns concerning the decomposition changes in carcasses. These studies are either observational or experimental, and both are directed to gather information such as insect checklists, environment temperature, and rainfall data of the study period, and the characteristics and duration of the decomposition stage [188].

Generally, observational-based successional studies are conducted to represent seasonal and ecosystem changes (terrestrial and aquatic), whereas experimental studies consider different death and carcass scenarios [189]. The data given in succession studies can be incorporated to help interpret the insect ecology and behaviour at a crime scene when geographical location and temperatures can be matched. In this regard, Matuszewski et al. (2019) [188] summarised global accounts of successional studies conducted using different carcass types under varied ecosystems.

### 4.4. Casework

Occasionally, a case is published which is the combination of facts and opinions of an FE who investigated a particular crime scene and may be relevant to the current case under investigation. Reference to casework can provide guidance to produce a robust witness statement but can also provide insight into areas where further research may benefit better interpretations. Some attempts have been made to describe how a forensic entomology report should be compiled with the latest methods by Kotzé et al. (2021) [190], detailing 16 subheadings which might be considered depending on the circumstances of a case.

## 5. Conclusions

This review essentially updates much of the current knowledge pertaining to forensic entomology, either described in the literature or experienced by the authors involved in casework. One of the major concerns of FEs is the education and practical training of proxies who attend crime scenes and collect insects and other biological materials. Many proxies may have basic skills in collecting and preserving insects across Gold, Silver, and Bronze standards and they can now pick and choose what they deem appropriate in respect to these activities. Reading this paper will help assist in their decisions. It cannot be emphasised enough that these activities need to be of a high quality if these evidence and associated information are to be of any benefit to the FE, and of course the case, especially for their ultimate consideration by the judiciary.

## Figures and Tables

**Figure 1 insects-14-00536-f001:**
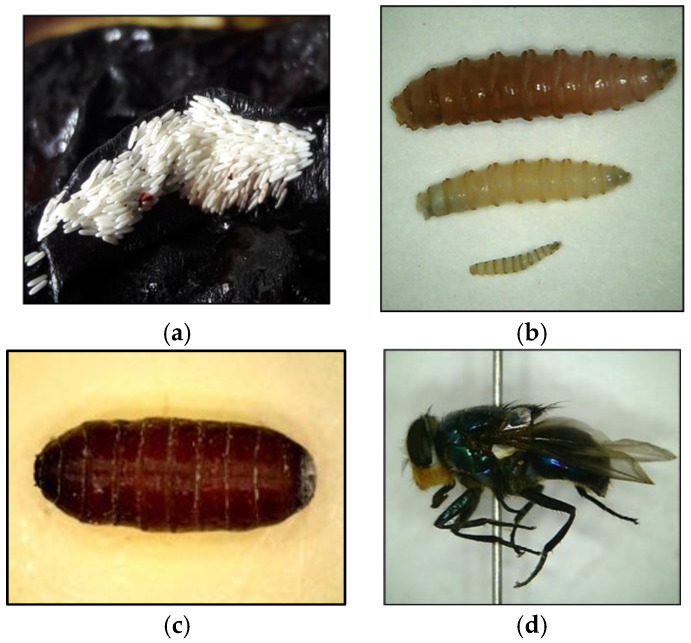
The different life stages of a fly. The measurements added are from Byrd and Tomberlin, 2020 [9]. (**a**) Fly egg raft (size range of single egg 1.2–2 mm). (**b**) Three larval instars (in ascending order; third, second, first instar (size range from 2–23 mm). (**c**) Fly pupa (size range from 6–23 mm). (**d**) Adult fly (size range from 6–14 mm).

**Table 1 insects-14-00536-t001:** A guide detailing three standards for forensic entomologists and their proxies to aid in collecting and preserving insect specimens.

Category	Specific Entomological Aspect	Gold Standard	Silver Standard	Bronze Standard
General	Person involved in sampling	Forensic entomologist (FE)	FE-trained police officer, medical examiner, or pathologist	Police officer or field technician (generally untrained)
	Number of times visiting a crime scene for and/or mortuary for entomology assessments	Initial visit followed by additional visits to gather climatic data and sift surrounding soil (especially when advanced decomposition is present)	Once	Once
	Clothing	Standard personal protection equipment ^1^ supplied by police/agency attending crime scene	Standard personal protection equipment ^1^ supplied by police/agency overseeing crime scene or person acting in the FE role	Masks and gloves minimum standard personal protection equipment ^1^ supplied by police/agency overseeing crime scene if present
*Microclimatic data*	Data collecting equipment	Onsite data collection—infrared thermometer, temperature probe, temperature data logging device; Offsite data collection—nearest meteorological station	Nearest meteorological station data, analogue thermometer	Nearest meteorological station data
	Type of data collected at a crime scene	Ambient temperature, larval mass temperature, soil temperature, humidity, photoperiod, rainfall from nearest weather station	Ambient temperature, larval mass temperature), rainfall from nearest weather station	Ambient temperature, rainfall from nearest weather station
	Duration of data collection	10–12 days at crime scene after discovery	Not applicable	Not applicable
*Sampling of insects*	Sampling equipment	Entomology kit ^2^, forceps (various sizes and types), spoons, and artist paint brushes, insect net, sticky traps, containers (closed and ventilated), refrigerated container (cooler or fridge) for transport	Disposable forceps, containers, preservatives (supplied by agency overseeing crime scene)	Disposable forceps, containers, preservatives (supplied by agency overseeing crime scene)
	Type and maturity stages of sampling insects	Eggs, larvae, full and empty puparia, adults of flies, and beetles	Eggs, larvae, full and empty puparia	Eggs, larvae, full and empty puparia
	Number of insects sampled	Depends on the location, access, and stage of life of the insects. The number of insects should cover the base of the container used	Depends on access, sample available insect material. Number of insects to cover the base of the container used. If possible, a minimum of 10 specimens chosen randomly	Depends on the access, sample available insect material (minimum 10 specimens chosen randomly)
	Labelling	Include date, time, case reference no., type of specimen via indelible pen on external label. Repeat in pencil and place this label inside container	Include date, time, case reference no., type of specimen via indelible pen on external label. Repeat in pencil and place this label inside container	Include date, time, case reference no., type of specimen using pencil on label both outside and inside container
*Preservation of insects* *for morphological analyses and minPMI*	Live insects (eggs/larvae/full puparia)	Retain a sample of insects (placed into ventilated containers) for later rearing. Place these in refrigeration or cooler. For eggs, add moist paper to avoid dehydration	Place these in refrigeration or cooler. If cooling device is unavailable, place all insect material into the preservative	Place these in refrigeration or cooler. If cooling device is unavailable, place all insect material into the preservative
	Hot water fix (larvae/full puparia)	Hot water taken to the field via a thermos flask or available at crime scene station set up by agency overseeing crime scene. Place larvae, pupae in hot water (1 min) before placing in preservative. Full puparia pierced, then placed in hot water, then in preservative	Obtain hot water from external source: e.g., take away food outlet. If no available hot water, then place insects directly into preservative (document preservative on label)	Obtain hot water from external source: e.g., take away food outlet or place insects directly into preservative (document preservative on label)
	Killing method (adults)	Spray bottle of ethanol to spray directly onto the insect collected in net then placed in 70% alcohol	Dipping insects caught in net into preservative and placed into 70% alcohol	Not applicable as adults; not collected
	Preservative (eggs/larvae/full and empty puparia) for minPMI	70–80% ethanol	Isopropyl alcohol, formalin or methanol ^3^ (document preservative on label)	Any white coloured alcohol (≥40% proof) obtained from a liquor outlet (document preservative on label)
*Preservation of insects* *for toxicological analyses*	Larvae, full and empty puparia	−20 °C	Freezer	Freezer
*Preservation of insects* *for molecular analyses*	Eggs, larvae, full and empty puparia, adults	100% ethanol	Freezer or if a preservative liquid is used (document preservative on label)	Freezer or if a preservative liquid is used (document preservative on label)
*Transportation*	Storage of insects for transporting	Refrigeration or cooler with frozen ice packs	If live insects retained, then place in a cooler with frozen ice packs	Not applicable all specimens preserved
*Preparation of insects for rearing*	Types of containers used for rearing	Plastic containers with feeding and bedding medium ^4^	Plastic containers provided by the police/agency at crime scene ^4^	Any sealable container; all specimens preserved
*Identification*	Method of identification	Morphological identification keys, molecular methods, hyperspectral and CT imaging, tomography, chemical methods	FE will conduct identification	FE will conduct identification
*Age determination*	Method of age determination	Larval instar, ADD method, isomegalen/isomorphen method, growth tables, crime scene resembling study within a growth chamber	FE will conduct age determination	FE will conduct age determination
*Expert verification*	Person involving in verification	Conducted by an alternate FE	Conducted by an alternate FE ^5^	Conducted by an alternate FE ^5^
	Findings needed to be verified	Species and age of the specimen	Conducted by FE ^5^	Conducted by FE ^5^
*Documentation*	Crime scene	Insect data collection form	Notebook	Notebook
	Decomposition stage	Determination based on carcass characteristics, photos and videos at crime scene and the mortuary. Previous experience and referring to existing publications	Send photographic and video evidence from crime scene to an FE	Send photographic and video evidence from crime scene to an FE

Notes: (1) e.g., closed-toed shoes, long pants, scrubs, and masks. (2) Entomology kit should contain all equipment necessary to collect and preserve insects collected: see [1,2,9,10]. (3) Formalin and methanol should be avoided if possible, but used if there is no other alternative. (4) Food (e.g., pork, beef) and bedding material (e.g., dry sand, vermiculite sawdust). (5) Police, crime scene technicians, and pathologist/medical examiners should have knowledge of their local FE.

**Table 2 insects-14-00536-t002:** Specific information that needs to be included into protocol sheets. Many review papers also list such protocols and should be consulted [1].

Type of Protocol	Category	Specific Information
Crime scene/autopsy	General case information	Date, case number, date and time of scene examination, name of the FE and investigation officer(s)
	Prior information from investigation officer (**IMPORTANT**)	Location found, date and time body recovered. Date when reported missing of the decedent and date and time last seen alive **SHOULD** only be available following the calculated minPMI to avoid bias in the analyses
	Recordings of the body	Position of body main axis, position of extremities, and position of head and face, location of body in reference to vegetation, and proximity to open doors, windows, or other openings if within a structure, description of clothing, type of debris on body, level exposure of the body (open air or burial; full sun or shade exposure), any detectable alterations to the body (i.e., natural, man-made, and scavenging marks)
	Recordings of the crime location (outdoor)	General habitat (rural or urban, terrestrial, or aquatic), sun or shade conditions, type of terrestrial ecosystem (1. rural; forest, tillable land, pasture, and crop, 2. urban; vacant lot, pavement, and rubbish container), type of aquatic ecosystem (lake, river, pond, irrigation canal, swamp, marshland), aquatic water type (fresh, brackish, and salt water)
	Recordings of the crime location (indoor)	Type of building (open or closed), doors and windows closed or open, on or off fans, lights, A/C and heaters
	Recordings of insect activity	Location of insect aggregations on the body, location of dispersing larvae, full and empty puparia up to 10 m away from the body. Attention should be given to the presence of ants
	Recordings at autopsy	Date and time of body placed into refrigeration after coming from the scene and removed prior to autopsy, locations of specimens (in/on body, clothing and other covers, body bag)
	Climate data (temperature)	Ambient temperature, larval mass temperature, temperature readings of A/C and heaters, moon phases
	Attachments	Diagram of body showing locations of insect mass infestations and sample locations, weather data from nearest meteorological station, photos of body in situ, autopsy photographs
Laboratory	Insect sampling	Sampled numbers (live and dead), type of insect (flies, beetles, or mites), growth stages (eggs, larvae, full and empty puparia, and adults)
	Sampling method	Fixation and preservation medium (% alcohol), date and time of preservation and fixation, sampled location (water, soil or arial)
	Identification	Identification method (morphological or molecular), specific species identified
	Age determination	Method of age determination, minPMI estimation based on available data

**Table 3 insects-14-00536-t003:** Development studies conducted on some forensically important fly species from different global regions. This table presents an overview only and is a starting point for any person interested in forensic entomology.

Species	Reference Studies with Study Locations
*Ch. megacephala*, *Ch. rufifacies*, *Ch. albiceps*, *L. sericata*, *L. cuprina*, *L. illustris*, *Calliphora vomitoria* (L.), *C. vicina*, *Phormia regina* (Meigen)	Bambaradeniya et al. (2023) (Asia, middle east, Australia, Europe) [23], Niederegger et al. (2010) (Germany) [156], Wang et al. (2016) (China) [157], Greenberg and Tantawi (1993) (USA) [158], Ames and Turner (2003) (UK) [159], Ireland and Turner (2006) (UK) [160], Niederegger et al. (2013) (Germany) [161], Wood et al. (2022) (UK) [162], Donovan et al. (2006) (UK) [163], Hwang and Turner (2009) (UK) [164], Aak et al. (2011) (Norway) [165], Sanei- Dehkordi et al. (2014) (Iran) [166], Baque et al. (2015) (Germany) [167], Byrd and Allen (2001) (USA) [168], Nabity et al. (2007) (USA) [169], Roe (2014) (USA) [170], Roe and Higley (2023) (USA) [171]
*Sarcophaga rufucornis* (Fabricius)	Sukhapanth et al. (1988) (Thailand) [172], Amoudi et al. (1994) (Saudi Arabia) [173], Nassu et al. (2014) (Brazil) [174], Adhikari et al. (2016) (India) [175], Bansode et al. (2016) (India) [176], Barbosa et al. (Brazil) [177]
*M. scalaris*	Prawirodisastro and Benjamin (1979) (USA) [178], Trumble and Pienkowski (1979) (USA) [179], Greenberg and Wells (1998) (USA) [180], Harrison and Cooper (2003) (USA) [181], Zuha et al. (2012) (Malaysia) [182], Feng and Liu (2014) (China) [136], Zuah and Omar (2014) (Malaysia) [183], Chakraborty et al. (2016) (India) [184], Thomas et al. (2016) (USA) [185], Ong et al. (2018) (Malaysia) [186], Castillo-Alanis et al. (2020) (Mexico) [187]

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
