# Peer review of "A Summary of Concepts, Procedures and Techniques Used by Forensic Entomologists and Proxies"

_insects, 2023, doi:10.3390/insects14060536_

Round 1

Reviewer 1 Report

Although I acknowledge the work the authors have put into this review: I do not see any benefit or new value in the manuscript, which mainly summarises what exists in other (cited) papers, even in summary form. The silver and bronze categories are more or less self-evident for the "non-specialists" and do not provide any significant new value. At the same time, I miss the discussion of new works and studies and also older but important publications on the topic of temperature modelling and storage/preservation. Some of the references are questionable, I would have cited other works. Furthermore, there are a few strange formatting and unnecessary spelling mistakes that are irritating. Moreover, individual paragraphs contain meaningless information (SEM methods in the determination of beetle families, etc.).

Author Response

How to answer this reviewer is difficult.  I can say, however that the reviewer should have read the paper as some of this information is new, especially summarising the information of how a proxy may go about sampling and preserving. Dadour has been teaching agencies such as the FBI and other Australian and global law enforcement agencies for 25 years about how to go about doing the FE’s job when an FE cannot attend a scene.  The question is always what can I do if I don’t have any equipment as specified, can I still retrieve useful information?  The inference from this reviewer is that they assume an FE always attends a case and this is not so.  Some of the references are questionable, how?  As it is we cite 190 references.  It seems that this reviewer is just using personal opinion to make such unjustified statement about meaningless information. Please forgive the text mistakes, I do hope we have sorted these out.

Author Response

Brief Summary: The authors present a review of practical information for both forensic entomologists and their proxies. Unlike previous, general reviews, this paper is focused on pertinent information/training for proxies so that a sufficient collection is still made for analysis and determination of some portion of the postmortem interval. The use of both a best practice (gold standard) and more field expedient methods (silver and bronze standards) is a helpful addition to both trainers and trainees.

I wish to thank reviewer 2 for the time spent on reading the document and adding some very useful comments.

General Concept Comments: A lot of information for proxies is provided in Table 1. This could be especially hard to follow and put into action for investigators with minimal training. Consider the addition of a working flow chart that could map back to Table 1.

This would be impossible to illustrate a work flow diagram as only an FE would classify themselves as a gold standard. The silver or bronze standard person may have possible components and skills to collect, analyse etc across all 3 standards.  Although on a rare occasion a gold standard person may have to compromise and adopt some of the silver or bronze standards. Instead in conclusion we have added such a comment.

Specific Comments: Keywords: Consider adding a keyword that reflects a main aim of the paper - additional collection/ documentation standards for proxies. Potentially, “insect collection guidelines” or “insect evidence collection.” Additionally, xenobiotic detection is comparatively a small portion of the review. Consider removing “xenobiotic” as a keyword.

This has been rectified in the text.

Line 50: Consider avoiding the term “direct clues” as this might be confused with direct (versus circumstantial) evidence.

We agree about direct clues.  A clue is a clue, and all clues are circumstantial unless they end up being used in the investigation and then they may become direct clues.

Lines 60-61: I believe that medical myiasis typically falls within the realm of medicolegal entomology. Supported by your citation (12) and use in the subsequent paragraph.

Added in text.

Line 90: As stated later in the paper, an expert can use an approach that is not matching with a isomegalen/isomorphen curve, like the use of ADH/ADD. I would include this here.

Corrected in text.

Lines 98-101: With the citation (27), Cleridae should be in this list. Additionally Kulshrestha and Satpathy (2001) discuss cases with dermestid and clerid evidence, but cite Entomology and Death (M.L. Goff, E.P. Catts, Arthropod Basics: Structure and Biology in: E.P. Catts, N.H. Haskell Eds.), Entomology and Death: A Procedural Guide, Joyce's Print Shop, Inc., Clemson, SC, 1990, pp. 46-47.) Also, I question if citation 9 is an appropriate citation for this sentence, a succession study(ies) would be more appropriate.

Cleridae has been added.  If you look up citation 9 you can easily track down all the necessary succession studies as it reviews them all.

Line 110: It is unclear to me, and likely other readers, why location-specific entomofauna would only be useful in early taphonomy and what exactly this time scale is. Is there a supportive statement/citation you could add? Additionally, a citation of the most recent review on the subject could be helpful to the reader. Charabidze, Damien, Matthias Gosselin, and Valéry Hedouin. "Use of necrophagous insects as evidence of cadaver relocation: myth or reality?." PeerJ 5 (2017): e3506. Lines 123-124: Citation (126) seems inappropriate. A more appropriate citation for this sentence would be something like: Baqué, Michèle, Jens Amendt, Marcel A. Verhoff, and Richard Zehner. "Descriptive analyses of differentially expressed genes during larval development of Calliphora vicina (Diptera: Calliphoridae)." International Journal of Legal Medicine 129 (2015): 891-902.

This supportive information has been added and references have been updated.

Line 278: “Pre-feeding” stages of a blow fly (as an example) would include the egg stage. I believe the others are referring to the “feeding” larval stages.

Rectified in text by removal of pre.

Lines 314-323: While decomposition stages can be informative, and a good tool for education/ discourse, a change in decomposition stage does not coincide with a discrete change in insect community (Schoenly and Reid 1987). This section should be reworded to avoid this association. Schoenly, K., and W. Reid. "Dynamics of heterotrophic succession in carrion arthropod assemblages: discrete seres or a continuum of change?." Oecologia 73 (1987): 192-202.

We do make the statement that “different decomposition stages can provide clues to determine the minPMI”.   This is written as such because it is not an absolute. Sometimes it can be useful, you really only need to refer to a pathologist or anthropologist who may well use discrete stages of decomposition to estimate the minPMI.

Lines 341-342: The cited study (72), did not investigate “microbial activities associated with the carcass decomposition.

A new reference more applicable has been added.

Lines 443-445: it is important to separate (with foil, smaller cup, etc.) the food source from the rearing media. Considering adding a sentence to make this clear.

This has been corrected in the text.

Line 495: I believe “antennaomeres” should be “antennomeres”

corrected

Line 650: should be Matuszewski

this has been corrected.

Figures and Tables

Figures 1-4 -These figures show the generalized stages in the life cycle of forensically important flies. Their reference in the text does not seem to aid reader understanding. If these figures were included for another reason, like to serve as a general guide to a proxy, this should be stated. If not, consider removal from the publication.

Table 1:

  • Consider adding categories that cover larval food and rearing media. These are touched on in the article but would be a helpful addition to the Table that organizes these items for proxies.

This is added in superscript in table 4

  • I believe the “Gold standard” for “Data collecting equipment” should read “data logging” not “data lodging”

Changed in text of table

  • For the duration of data collection, I believe the “Silver standard” should be “Not applicable” if the number of times visiting the crime scene is “once."

Changed in text of table

  • Some citations in the Table are necessary, or at a minimum, need further explanation/ support in the text. For example
    • a citation for “10-12 days” for microclimatic data collection, collection of a minimum of 10 specimens,
    • and recording pupae up to 10m away from the body (Table 2).

I disagree, everything is in the text under similar headings.  Referencing each and every point would be repetitious and it distracts from the table

  • For sampling equipment, data suggests that sticky traps are not the gold standard for collection at a scene (Sanford 2017). Sanford, Michelle R. "Comparing species composition of passive trapping of adult flies with larval collections from the body during scene-based medicolegal death investigations." Insects 8, no. 2 (2017): 36.

In Dadour’s experience sticky traps definitely are useful inside dwellings.  I have trapped the adult flies corresponding to larval stages at 5 crime scenes.  They are Gold standard as a just in case piece of kit.

  • Potentially an alcohol resistant pen or marker should be the “Gold standard” instead of a Sharpie or similar permanent marker. I find that the ink from a Sharpie will still bleed a bit if preservative spills over onto the external label. - In a pinch, the “Bronze standard” for labelling could be to use a pencil for both external and internal labels.

We have changed sharpie to indelible pen. Good point you raised about pencils for bronze standard, and we have corrected for this.

  • If live insects samples can’t be cooled, why not have a lower standard be to record the temperatures the live insects were exposed to and start the rearing at the scene, instead of simply not taking a live sample? I would argue that this could still be helpful, especially in identification of the reared adults.

Rearing at scene for proxies is difficult so we disagree.  Silver and Bronze standards are never going to achieve this.  Dadour, personal experience of attending over 140 scenes, the police in Australia run very controlled crime scenes and generally want the entomology to be completed as quick as possible and as refrigeration is available (crime scene trucks are equipped with fridges), then this is the best solution between scene and laboratory.

  • Why not consider a preservative like KAA in place of hot water? Especially if pre-filled vials are included in collection kits.

KAA is great for preventing discolouration, but all samples are required to be placed into alcohol for storage as soon as possible. KAA does cause minimal size. Normal practice states it is always best to hot water kill.  BTW hot water is very available unlike KAA.

  • It is unclear why the Bronze standard for the “Killing method (adults)” is “Not applicable.” If this section is only for sweep net collection of adults and this is not performed in the Bronze standard, please make this clear.

This is now clear in table

  • Should the “Preservative (eggs/larvae/pupae)” also include adults?

This section has been expanded and covers this aspect as well as the number of ways to preserve now in table for morphology, toxicology and DNA.

Table 2: -

  • Consider adding ambient temperatures for both shaded and unshaded areas in the “Climate data” category. Especially if the remains are in partial sun/shade at any time of day.

From our own perspective this is generally not possible as the FE is present at a single point in time.  The person attending at any standard needs to make that judgement of whether in shade or full sun.  The FE may return to the scene if allowed and could measure this, but for cadavers that have been exposed for long periods of time, it becomes impossible to measure the past history of partial sun/shade exposure. This has been added to the table.

  • Consider adding type of preservative to the “sampling method” category.

This has been added in table .

Table 3: -

  • This table is not exhaustive for forensically important species or reference studies for each species (especially for the selected blow fly species). I think it is pertinent to add why species/studies were selected. For example, was a specific range of years used?

We state in table caption now that this is not an exhaustive list but these can be used as a starting point for anyone interested in becoming an FE

  • For the blow fly reference studies, it would be helpful to section them, so they could be associated with a species.
  • this is difficult as some of these studies considered more than 1 species at the same time.

L_u_c_i_l_i_a_ _s_e_r_i_c_a_t_a_ _should be L_._ _s_e_r_i_c_a_t_a_ _if the previous form for C_h_r_y_s_o_m_y_a_ _is followed. - “M_e_g_a_s_e_l_i_a_._ _S_c_a_l_a_r_i_s_” _should be “M_e_g_a_s_e_l_i_a_ _s_c_a_l_a_r_i_s_”.

this has been corrected.

References Citation #57 - More information is needed. I believe this citation is from a conference presentation, but that information is not included.

When referencing an app in a scientific paper, we have now used the American Psychological Association style:

Magni, P. A.; Dadour, I. SmartInsects – Forensic Entomology. 2013. (Version 1.0). Mobile application software: iOS. Retrieved from App Store, URL: https://apps.apple.com/us/app/smartinsects/id985074731

Reviewer 3 Report

This a review paper so there is no research to analyze. The paper is clear and well written and puts together much information that was scattered here and there. I see no obvious places for change of the information or the writing. Some of the photos are not as clean and crisp as they might be but they serve the purpose. 

Author Response

Thank you for your appraisal of the MS.  It did require some editing as 3 reviewers made comments.  Resolution of photos are always difficult digitally  

Round 2

Reviewer 1 Report

Thank you for your response. However, I stay with my opinion. And that's just because I actually read the paper, but thanks for the authors' kind recommendation to read it.

Authors response:

"The inference from this reviewer is that they assume an FE always attends a case and this is not so"

Hmm, no, I did not say that - I wrote: The silver and bronze categories are more or less self-evident for the "non-specialists" and do not provide any significant new value.

Authors response:

"Dadour has been teaching agencies such as the FBI and other Australian and global law enforcement agencies for 25 years about how to go about doing the FE’s job when an FE cannot attend a scene"

I did it too - and I disagree with Dadour.

Authors response:

"Some of the references are questionable, how?  As it is we cite 190 references."

Well, I don't think that the number of references says anything about their quality? As you e.g. can see by the recommendations and now new included references.

Such a scatter-shot paper with notes on preservation (whose usefulness I doubt beyond what already exists in the literature), literature on development as in Table 3 (still missing important citations despite impressive 190 references), literature on insect development as in Table 3 (which still lacks important citations, despite an impressive 190 references), explanations on identification and identification literature, succession studies, what else is this paper supposed to do?

I also doubt the structure of the manuscript, I am sorry.  As I said, I acknowledge the work behind it, but I'm not convinced by the quality of the manuscript - I don't see anything beyond the existing literature and guidance that helps improve sampling/preservation.